

# Development of a circHIPK3-based ceRNA network and identification of mRNA signature in breast cancer patients harboring BRCA mutation

Qi-xin Lian[1,*], Yang Song[2,*], Lili Han[3], Zunxian Wang[4] and Yinhui Song[5]

[1] Oncology Department, First Affiliated Hospital of Jiamusi University, Jiamusi, China
[2] Pathology Department, First Affiliated Hospital of Jiamusi University, Jiamusi, China
[3] General Surgery, The Seventh Medical Center of the General Hospital of the Chinese People's Liberation, Jiamusi, China
[4] Chemoradiotherapy, First Affiliated Hospital of Jiamusi University, Jiamusi, China
[5] Department of Breast Surgery, The First Hospital of Qiqihar, The Affiliate Qiqihar Hospital of Southern Medical University, Qiqihar, China
[*] These authors contributed equally to this work.

Corresponding author
Yinhui Song, yhuisong@126.com

## ABSTRACT

**Background**. Exploring the regulatory network of competing endogenous RNAs (ceRNAs) as hallmarks for breast cancer development has great significance and could provide therapeutic targets. An mRNA signature predictive of prognosis and therapy response in BRCA carriers was developed according to circular RNA homeodomain-interacting protein kinase 3 (circHIPK3)-based ceRNA network.

**Method**. We constructed a circHIPK3-based ceRNA network based on GSE173766 dataset and identified potential mRNAs that were associated with BRCA mutation patients within this ceRNA network. A total of 11 prognostic mRNAs and a risk model were identified and developed by univariate Cox regression analysis and the LASSO regression analysis as well as stepAIC method. Genomic landscape was treated by mutect2 and fisher. Immune characteristics was analyzed by ESTIMATE, MCP-counter. TIDE analysis was conducted to predict immunotherapy. The clinical treatment outcomes of BRCA mutation patients were assessed using a nomogram. The proliferation, migration and invasion in breast cancer cell lines were examined using CCK8 assay and transwell assay.

**Result**. We found 241 mRNAs within the circHIPK3-based ceRNA network. An 11 mRNA-based signature was identified for prognostic model construction. High risk patients exhibited dismal prognosis, low response to immunotherapy, less immune cell infiltration and tumor mutation burden (TMB). High-risk patients were sensitive to six anti-tumor drugs, while low-risk patient were sensitive to 47 drugs. The risk score was the most effective on evaluating patients' survival. The robustness and good prediction performance were validated in The Cancer Genome Atlas (TCGA) dataset and immunotherapy datasets, respectively. In addition, circHIPK3 mRNA level was upregulated, and promoted cell viability, migration and invasion in breast cancer cell lines.

**Conclusion**. The current study could improve the understanding of mRNAs in relation to BRCA mutation and pave the way to develop mRNA-based therapeutic targets for breast cancer patients with BRCA mutation.

## INTRODUCTION

Breast cancer ranks the first in the incidence of female cancers and occupies the fifth place of cancer-related mortality, accounting for more than 2 million new cases and 685,000 deaths (*Sung et al., 2021*). For breast cancer, heritage as a critical risk factor has been reported that genetic proportion of familial clustering breast cancer is approximately 73% (*Wendt & Margolin, 2019*). Certain genomic mutations such as BRCA2 and BRCA1 participate in breast cancer development (*Yoshimura, Imoto & Iwata, 2022*). Patients harboring germline BRCA mutation are initially diagnosed as advanced breast cancer and it is associated with more unfavorable outcomes (*Ho et al., 2021*). Although in healthy individuals carrying BRCA1/2, bilateral mastectomy can reduce breast cancer risk, the overall survival for most BRCA1/2 carriers remain unclear (*Peleg Hasson, Menes & Sonnenblick, 2020*). Additionally, radiotherapy and platinum-based chemotherapy need personalized application in breast cancer patients with BRCA1/2 (*Peleg Hasson, Menes & Sonnenblick, 2020*). Current markers do not accurately define the personalized treatment of breast cancer and the emergence of drug resistance during breast cancer drug therapy. Therefore, it is necessary to develop novel biomarkers in the clinical management of BRCA carriers.

As a class of noncoding RNA, circular RNA (circRNA) is involved in gene regulation. Accumulating evidences have demonstrated that circRNAs are implicated in pathological conditions and contribute to the pathogenesis of diverse diseases like cancers, neurological diseases, immune diseases, cardiovascular diseases (*Han, Chao & Yao, 2018*). CircHIPK3 derives from exon 2 from the HIPK3 gene, which exerts oncogenic or tumor suppressive effects through serving as microRNA (miRNA) sponges (*Xie et al., 2020*). CircHIPK3 could promote tumor growth of glioma, prostate cancer, epithelial ovarian cancer, BCRA and colorectal cancer (*Cai et al., 2019*; *Chen et al., 2020*; *Jin et al., 2018*; *Liu et al., 2018*; *Zeng et al., 2018*), while it suppresses bladder cancer progression (*Li et al., 2017*). Targeting circHIPK3 is a promising therapeutic approach to the management of cancer. However, the role of circHIPK3-related genes in breast cancer carrying BRCA1/2 has not been fully elucidated. Hypothesis of ceRNAs represents a large-scale regulatory network that explains the communications of long noncoding RNAs (lncRNAs), circRNAs, messenger RNAs (mRNAs), and pseudogene RNAs with miRNAs mediated by miRNAs response elements. Dysfunction of ceRNA network contributes to development of cancers, while it may provide opportunities for cancer treatments (*Salmena et al., 2011*).

In this study, we constructed a circRNA-miRNA-mRNA ceRNA network on the basis of circHIPK3 and identified potential mRNAs that were associated with BRCA mutation patients. A total of 11 mRNAs were filtered by univariate Cox regression analysis and LASSO regression analysis for prognostic model development. Clinicopathologic features, immune characteristics, genomic landscape, and pathway characteristics were analyzed between the risk groups to understand the potential mechanism of an 11 mRNA-based

signature in BRCA. The responsiveness to chemotherapy and immunotherapy was further predicted to offer a direction for personalized therapies in BRCA mutation patients.

## METHODS

### Data collection and pre-processing

The clinical follow-up and expression data of breast cancer patients with BRAC1/2 susceptibility genes BRCA were collected from TCGA database (https://portal.gdc.cancer.gov/), which included RNA-Seq expression profiles and single nucleotide variants (SNVs) processed by mutect2. Samples living longer than 30 days were retained after eliminating those without survival time or follow-up information. We converted ensemble gene IDs to gene symbol IDs, and the gene with multiple gene symbols was expressed as median value.

We downloaded GSE173766 dataset containing circRNA and mRNA sequencing data from GEO (https://www.ncbi.nlm.nih.gov/geo/) database. Moreover, we obtained the gene expression profiles of BRCA mutation samples in GSE103091, GSE16446, GSE20685, GSE20713, GSE42568, and GSE48390 from GEO database. RMA function in "affy" package (*Gautier et al., 2004*) was used to process each GSE dataset, followed by removal of batch effects using removeBatchEffect function embedded in "limma" package (*Ritchie et al., 2015*) was utilized to remove. We converted the probe to gene symbol, and removed normal tissue samples. Patients lacking overall survival (OS) and status, follow-up information were excluded, while samples survived longer than 30 days were kept. The batch effect was presented using t-distributed stochastic neighbor embedding (TSNE) (Fig. S1) based on the expression profiles of GSE dataset before and after the removal.

### Construction of circHIPK3-based ceRNA network

We obtained hsa_circ_0021603 corresponding gene HIPK3 from the circRNA data in the GSE173766 dataset, and then calculated the correlation of hsa_circ_0021603 with mRNA using Pearson. Positive correlated genes were selected under the threshold of Corr >0.8 and $p < 0.05$. Subsequently, we predicted hsa_circ_0021603 target miRNAs on the circBank website (http://www.circbank.cn/index.html). Meanwhile, we predicted target genes of miRNAs using miRtarbase, mircode, microT, TargetMiner, TargetScan, miRDB, miRanda, miRmap, starbase, PITA, RNA22, and PicTar. We reserved miRNA-mRNA in at least two databases, and these mRNAs were significantly positively correlated with hsa_circ_0021603. A ceRNA network was constructed using mRNAs, circRNA, and miRNAs.

### Functional enrichment analysis

GO analysis on biological process (BP), molecular function (MF), and cellular component (CC) categories as well as KEGG analyses using "WebGestaltR" package (*Liao et al., 2019*) in R were conducted for functional enrichment analysis.

### Developing and validating an mRNA prognostic model

In the ceRNA network, 241 mRNAs were used for additional study. We partitioned the GSE dataset at random into Train and Test datasets with a 1:1 ratio. The Train dataset was used to identify prognostic genes with P 0.05 using univariate Cox regression analysis,

followed by LASSO regression analysis to refine the model using "glmnet" package (*Hastie, Qian & Tay, 2021*). Finally, key prognostic genes were determined by stepwise multivariate regression analysis with stepAIC.

The risk score was calculated in accordance with the formula as follows:

Risk score $= \sum \beta \text{i*ExPi}$, where i indicated the gene expression levels of prognosis related genes, and $\beta$ represented Cox regression coefficient. Through surv_cutpoint function in "survminer" package (*Kassambara et al., 2017*), the optimal cutoff was determined and patients in Train dataset were grouped into low-risk and high-risk groups. Kaplan–Meier curves for each risk group were plotted and the significance of differences was defined by log-rank test. Besides, we conducted receiver operating characteristic (ROC) analysis using "timeROC" package (*Blanche, 2015*) and evaluated the areas under the ROC curve (AUCs) for 1-year, 3-years, 5-years and 7-years. To validate the robustness of predictive risk model, we calculated the risk score and predicted survival probability in TCGA cohort.

## Analysis of clinicopathologic features and SNV between risk groups

We analyzed the differences in human epidermal growth factor receptor-2 (Her2), progesterone receptor (PR), N Stage, estrogen receptor (ER), T Stage between high-risk groups using Fisher's exact test. Differences in TMB in the TCGA dataset were further investigated. In mutation dataset from TCGA dataset, we screened 8,707 genes with mutation frequency > 3 and screened significant somatic mutation between risk groups using Fisher's exact test with $P < 0.05$. Association between risk score and TMB was assessed with Pearson correlation analysis. Additionally, risk score distribution in different clinicopathologic features (age, ER, PR, and Her2, T Stage, N Stage, Stage, M stage). Wilcox tests were used for the analysis of differences between two groups, and the Kruskal-Wallis test assessed group differences.

## Gene Set Enrichment Analysis (GSEA) analysis between risk groups

Thereafter, we analyzed biological processes between risk groups through GSEA function in "clusterProfiler" package in TCGA dataset and GSE datasets using HALLMARK pathways in h.all.v7.4.symbols.gmt. Candidates were considered as significant enriched pathways under false discovery rate (FDR) < 0.25 and $P < 0.05$.

## Immune characteristics and pathway characteristics differences between risk groups

The scores of 28 immune cells from previous study (*Charoentong et al., 2017*) were calculated using single sample GSEA (ssGSEA) method between risk groups in GSE datasets. The immune score was evaluated using ESTIMATE algorithm. Additionally, MCP-counter was employed to assess immune score of 10 immune cells. From previous study (*Danilova et al., 2019*), we obtained several immune checkpoint genes and compared their distribution in risk groups. In order to study the pathways characteristics of risk groups, we scored the pathways in h.all.v7.4.symbols.gmt and human gene signatures in previous published research (*Mariathasan et al., 2018*) using ssGSEA and analyzed their correlation with risk score in TCGA dataset.

## Response to chemotherapy and immunotherapy of risk groups

The tumor immune dysfunction exclusion (TIDE) algorithm (http://tide.dfci.harvard.edu/) evaluates the M2 subtype of tumor-associated macrophages (exclusion), myeloid-derived suppressor cells, tumor-associated fibroblasts that limit the infiltration of T cells in tumors. Furthermore, two different mechanisms of tumor immune escape, including the dysfunction score of tumor-infiltrating cytotoxic T lymphocytes (dysfunction) and rejection score of CTLs by immunosuppressive factors is also studied with TIDE. A high TIDE score denotes a low rate of ICI therapeutic response. Response prediction to immune checkpoint inhibition (ICI) therapy was performed using TIDE. Additionally, we used the "pRRophetic" package (*Geeleher, Cox & Huang, 2014*) to calculate the half-maximal inhibitory concentration (IC50) in risk groups both in TCGA and GSE datasets.

## Performance of predictive model in immunotherapy datasets

IMvigor210 (*Balar et al., 2017*), GSE135222 (*Kim, Choi & Jung, 2020*), and GSE78220 (*Hugo et al., 2016*) are immunotherapy datasets. Risk score in the above three immunotherapy datasets were calculated and then survival chances were predicted using Kaplan–Meier curves with AUCs. Meanwhile, we used Fisher's exact test to compare the clinical response distribution amongst risk groups, including partial response (PR), complete response (CR), progressive disease (PD), and stable disease (SD). Risk score distribution in different clinical response groups was compared using Kruskal-Wallis test or wilcox.tests.

## Construction of nomogram

In TCGA dataset, a decision tree based on stage, age, T stage, risk type, N stage, M stage was produced to determine risk subtypes and we generated Kaplan–Meier curves of risk subtypes. The significant differences were determined by log-rank test. Univariate and multivariate Cox regression analysis were further used to select predictor of prognosis. Clinical outcomes were evaluated by a nomogram constructed using risk type and clinicopathological features (stage and age, T stage, M stage, N stage). The accuracy and reliability of predictive risk model were analyzed with calibration curves and decision curve analysis (DCA).

## Cell culture and transient transfection

MCF-10A, MDA-MB-231, MCF-7 cells were commercially bought from ATCC (Manassas, USA). The cells were cultured in Gibco DEME F-12 medium (Thermo Fisher, Scientific, Waltham, MA, USA). The negative control (NC) and circHIPK3 siRNA (Invitrogen, Carlsbad, CA, USA) were transfected into the cells utilizing Lipofectamine 2000 (Invitrogen, Carlsbad, CA, USA). The target sequences for circHIPK3 siRNAs were ACCATATGTTTATCAAACTCAGT (circHIPK3-si).

## Quantitative real-time polymerase chain reaction (qRT-PCR)

TRIzol reagent (Sigma-Aldrich, USA) was applied to extract total RNA from breast cancer cells. The HiScript II Q RT SuperMix (Vazyme, Beijing, China) was used to reverse-transcribe 500 ng of RNA into cDNA. The SYBR Green Master Mix (Thermo

Fisher, Scientific, Waltham, MA, USA) was used for qRT-PCR in an ABI 7500 Fast System. For a total of 45 cycles, the PCR amplification settings were at 94 °C for 10 min (min), at 94 °C for 10 s (s), and at 60 °C for 45 s. The internal reference was GAPDH. The primer sequences of GADPH were F: 5′-AATGGGCAGCCGTTAGGAAA-3′and R: 5′-GCCCAATACGACCAAATCAGAG-3′. The primer sequences of circHIPK3 were F: 5′-TTCAACATGTCTACAATCTCGGT-3′and R: 5′-ACCATTCACATAGGTCCGT -3′.

## Cell viability

The Cell Counting Kit-8 assay (Beyotime, Jiangsu, China) was performed according to the manufacturer's protocol for cell viability detection. Different treatment groups of cells were cultured at a density of $1 \times 10^3$ cells per well in 96-well plates. CCK-8 solution was applied at indicated time points. After incubating the cells at 37 °C for 2 h, the OD 450 values of each well were detected using a microplate reader.

## Transwell assay

Transwell assays for migration and invasion of MDA-MB-231 and MCF-7 were performed. Briefly, in a nutshell, Matrigel (BD Biosciences, Franklin Lakes, NJ, USA) was coated or left uncoated in chambers before cells (5 104) were added (for migration). In the upper layer, serum-free medium was supplied, and in the lower layer, full DMEM medium. Following a 24-hour incubation period, migrating or invading cells were fixed by 4% paraformaldehyde and stained with 0.1% crystalline violet, counted under a microscope.

## Statistical analysis

R software (version 4.0.4) was used in all the statistical analyses, with a two-tailed $P < 0.05$ being considered as statistical significance.

# RESULTS

## Construction of a circHIPK3-based ceRNA network

With the threshold of Corr > 0.8 and $P < 0.05$, we screened 1,358 mRNAs that were positively correlated with hsa_ circ_ 0021603. Subsequently, we predicted 45 hsa_ circ_ 0021603 target miRNAs. Through predicting target genes of 45 miRNAs, we reserved miRNA-mRNA in at least two databases, and these mRNAs were significantly positively correlated with hsa_circ_0021603. Finally, a ceRNA network containing 1 circRNA, 40 miRNAs, and 241 mRNAs was constructed (Fig. 1A). Based on functional enrichment analysis of 241 mRNAs, we found that these mRNAs were closely related with breast cancer, ErbB signaling pathway, Ras signaling pathway, and pathways in cancer (Figs. 1B–1E). The findings suggested that circHIPK3 competing with related mRNA for miRNAs, which may play a regulatory role in the occurrence of cancer.

## Development and verification of mRNA prognostic model

According to the ceRNA network, 241 mRNAs that were closely related to breast cancer were used for screening mRNAs for prognostic model construction. Through univariate Cox regression analysis, 40 mRNAs associated with prognosis were screened in the Train dataset (Fig. S2A). LASSO Cox regression analysis was employ to further refine the model.

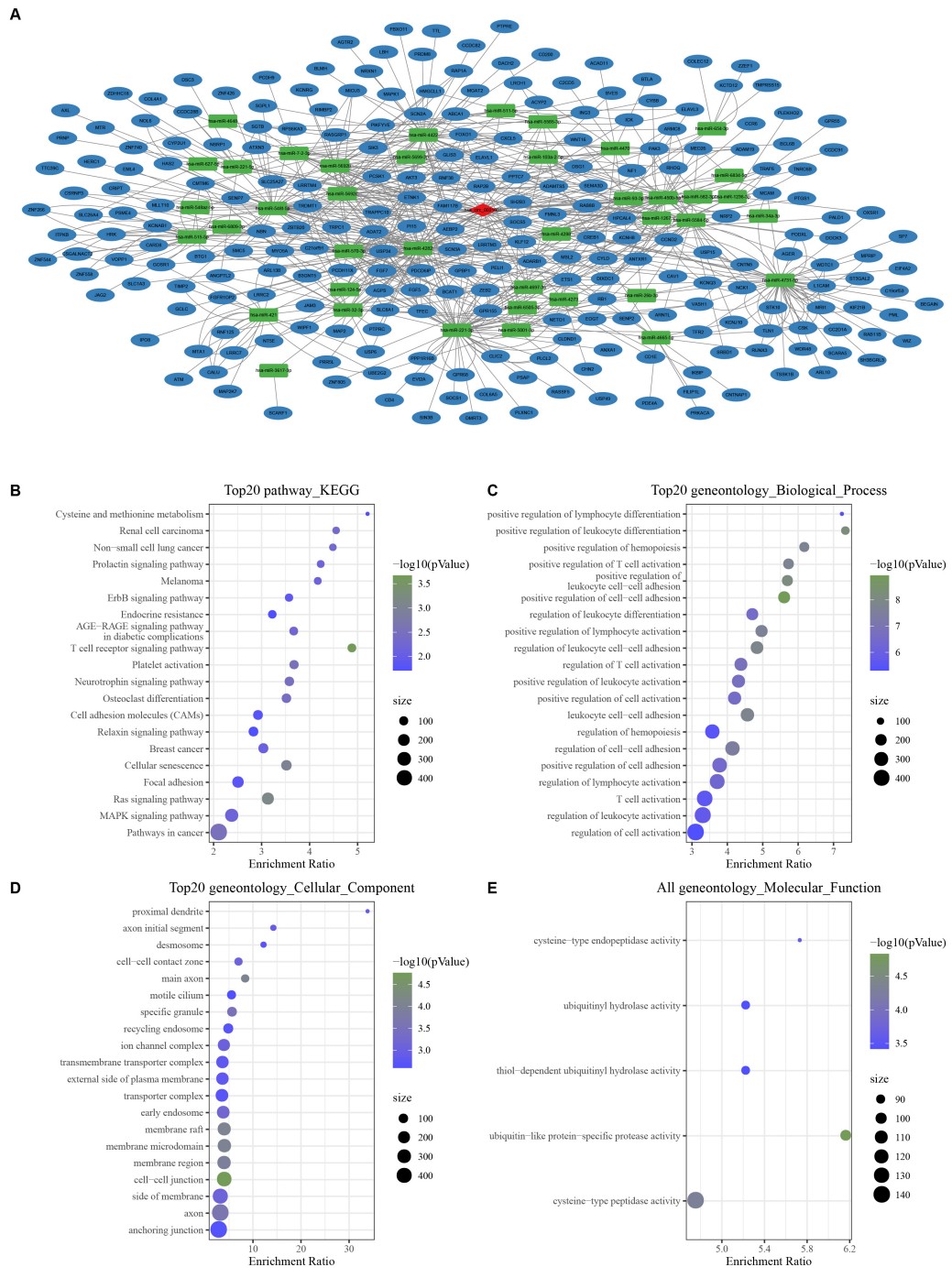

**Figure 1 Construction of a circHIPK3-based ceRNA network.** (A) Visualization of circRNA-miRNA-mRNA network. For 241 mRNAs in the network: (B) Top 20 KEGG enrichment analysis. (C) Top 20 BP enrichment analysis. (D) Top 20 CC enrichment analysis. (E) MF enrichment analysis.

Independent variable coefficients that are progressively heading to zero rose as lambda was steadily increased (Fig. S2B). Figure S2C displays the confidence interval under each lambda from 10-fold cross-validation. 21 mRNAs were found when lambda = 0.0182. With stepwise multivariate regression analysis and stepAIC, a total of 11 mRNAs were identified, including four risk mRNAs and seven protective mRNAs (Fig. S2D).

The risk score was calculated of each patient based on 11 mRNAs (Fig. 2A) as follows: riskScore = 0.652*ANGPTL2−0.289*CD1E−0.282*COLEC12−0.582*FAM117B+0.314* JAG2+0.186*L1CAM−0.405*MSL2+0.718*PSME4−0.42*SH2B3−0.31*TTC39C−0.308* ZBTB20. Survival analysis showed high risk patients exhibited poor prognosis compared with low risk patients ($P < 0.0001$) with 1-year AUC of 0.67, 3-year AUC of 0.75, 5-year AUC of 0.77 and 7-year AUC of 0.77 in the Train dataset (Fig. 2B). Similar results were also found in the Test dataset and GSE datasets, respectively (Figs. 2C–2D). We further validated the robustness of this model in TCGA dataset. Figure 2E revealed that high risk patients had lower OS ($P < 0.0001$), progression-free interval (PFI), disease-free interval (DFI) ($P = 0.0034$), disease specific survival (DSS) ($P < 0.0001$) ($P < 0.0001$) than that of low risk patients.

## GSEA analysis between risk groups

In GSE and TCGA datasets, EPITHELIAL_MESENCHYMAL_TRANSITION, GLYCOLYSIS, HYPOXIA were significantly enriched in high risk group, while ESTROGEN_RESPONSE, BILE_ACID_METABOLISM, FATTY_ACID_METABOLISM, were significantly enriched in low risk group (Figs. 3A–3B).

## Analysis of clinicopathologic features and SNV between risk groups

Thereafter, we assessed the distribution of risk score in N stage, T stage, ER, PR, and Her2, stage. Patients with T3 stage, N2 stage, stage, ER negative, PR negative and Her2 positive status had higher risk score (Fig. 4A). As shown in Fig. 4B, we found that high risk patients exhibited higher proportion of T2 stage, N2 stage, positive ER, negative PR, and positive Her2. We further evaluated the differences in genomic changes between risk groups. High risk patients possessed elevated TMB ($P < 0.0001$) (Fig. 4C) and had a positive correlation between risk score and TMB ($P = 4.18e−09$, $R = 0.191$) (Fig. 4D). In mutation dataset from TCGA dataset, we screened 612 significant mutated genes and displayed the top 20 in Fig. 4E. TP53 (33.73%), PIK3CA (33.3%), CDH1 (14.45%) were the most frequently mutated genes.

## Immune characteristics between risk groups

Furthermore, we evaluated the immune landscape in high and low risk groups in GSE datasets. We found that effector memory CD8 T cell, Type 1 T helper cell, activated B cell, effector memory CD4 T cell, activated CD4 T cell, eosinophil, activated CD4 T cell, natural killer cell, CD56 bright natural killer cell were significantly increased in low risk patients (Fig. 5A). Besides, low patients had higher adaptive score and innate score (Fig. 5B), StromalScore, ImmuneScore, and ESTIMATEScore were higher than that of high risk patients (Fig. 5C). Figure 5D deciphered that NK cells, cytotoxic lymphocytes and T cells were distinctly infiltrated in tumors of low patients. Several immune checkpoints such

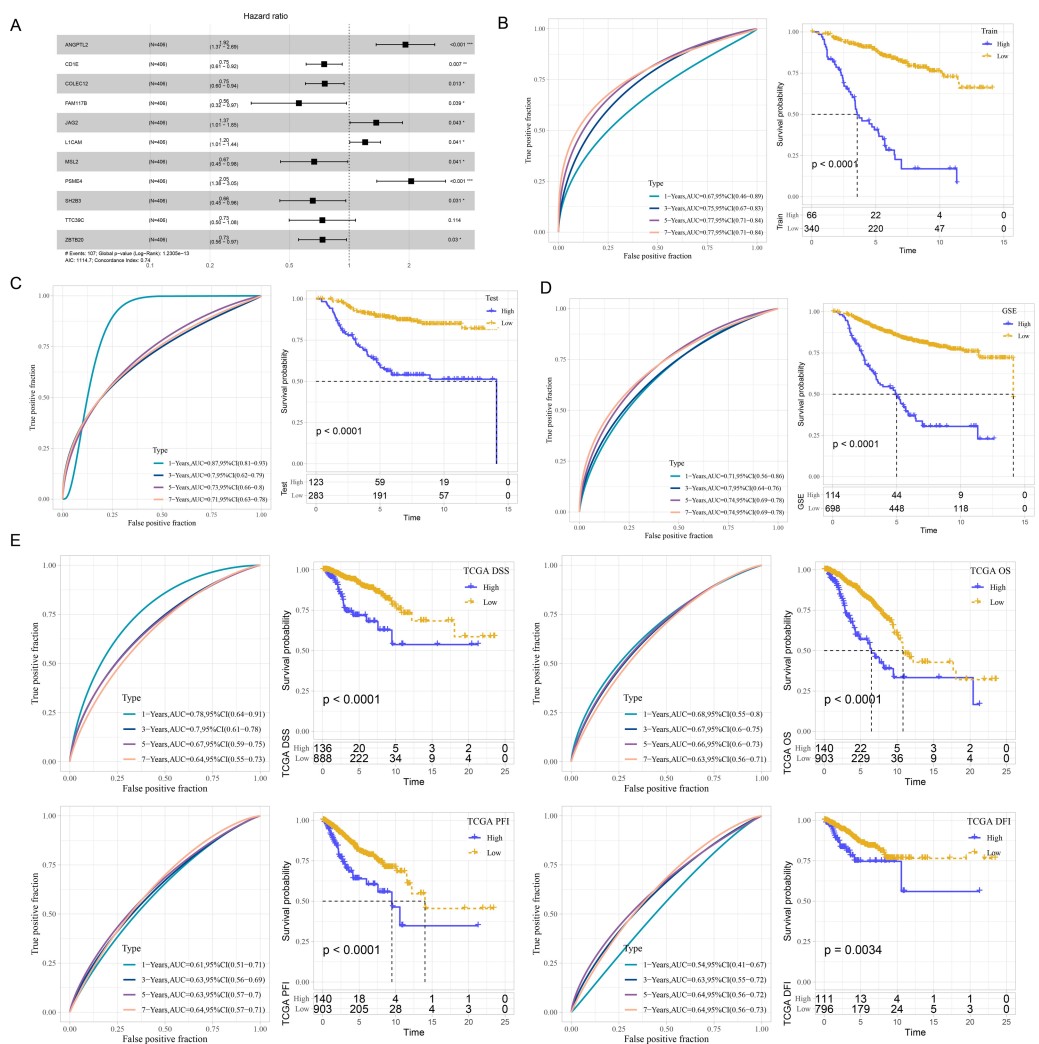

**Figure 2  Development and verification of mRNA prognostic model.** (A) Multivariate Cox analysis for model genes. (B) ROC curves for 1-year, 3-years, 5-years and 7-years and Kaplan–Meier curves of risk score in the Train dataset. (C) ROC curves for 1-year, 3-years, 5-years and 7-years and Kaplan–Meier curves of risk score in the Test dataset. (D) ROC curves for 1-year, 3-years, 5-years and 7-years and Kaplan–Meier curves of risk score in GSE datasets. (E) Validation of predictive efficiency in TCGA dataset.

as BTLA, CD200, CD200R1, CD274, CD28, ICOS, IDO2, LAG3 and PDCD1LG2 were overexpressed in low risk patients (Fig. 5E).

## Relationship of risk score and pathway characteristics

Meanwhile, we compared the distribution of pathways in TCGA dataset and found that there were 29 significant altered pathways in low and high risk groups (Fig. 6A). Association of risk score with 29 significant altered pathways was further analyzed (Fig. 6B). GLYCOLYSIS, EPITHELIAL_MESENCHYMAL_TRANSITION, and HYPOXIA were dramatically positively correlated with the Risk score. Additionally, risk score was in a

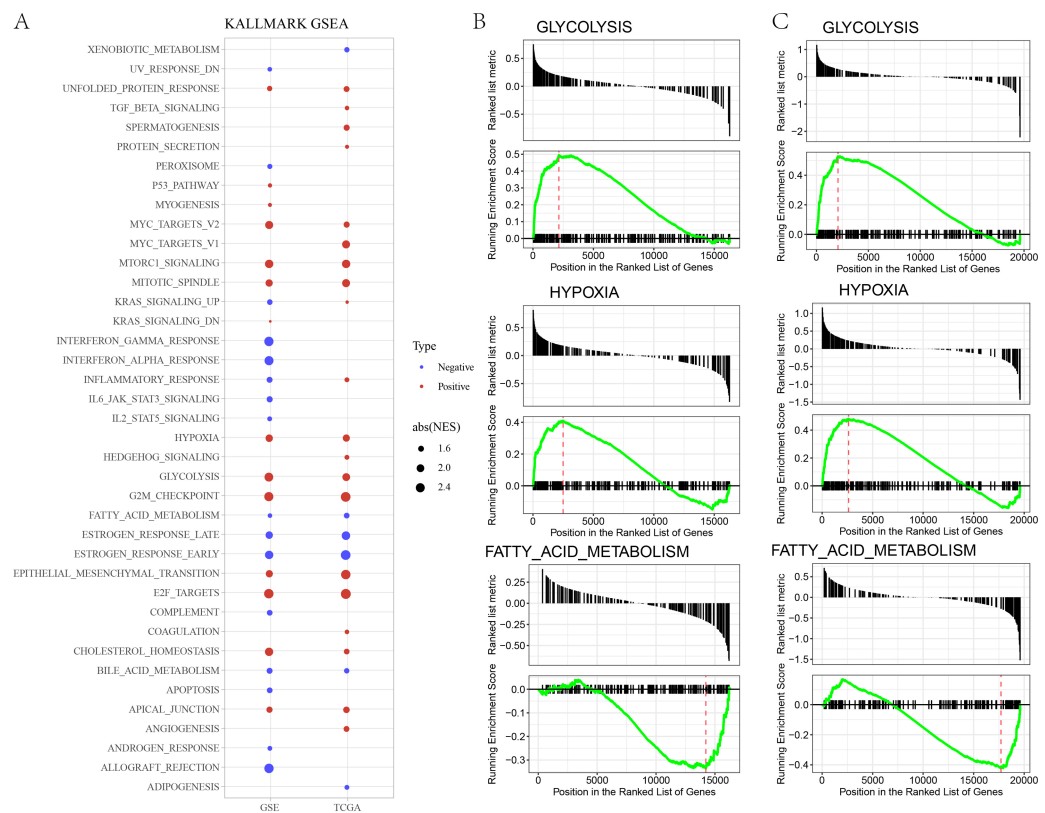

**Figure 3** (A) GSEA analysis between risk groups in TCGA and GSE datasets. (B–C) Representative results of GSEA analysis in GSE datasets and TCGA dataset.

positive relationship to DDR, cell cycle, DNA replication, homologous recombination, mismatch Repair (Fig. 6C).

## Prediction of response to immunotherapy and chemotherapy of risk groups

Through TIDE algorithm, we observed low risk patients exhibited lower TIDE score, indicating a high response to ICI therapy, while high risk patients had higher exclusion and MDSC both in TCGA and GSE datasets (Figs. 6D–6E). Additionally, low risk patient were sensitive to 14 drugs and high risk patients were more sensitive to 29 anti-tumor drugs in TCGA (Fig. 6F). Low risk patient were sensitive to 47 drugs and high risk patients were sensitive to six anti-tumor drugs in GSE (Fig. 6G).

## Performance of predictive model in immunotherapy datasets

The performance of risk model was verified in three immunotherapy datasets. High risk patients had unfavorable prognosis in IMvigor210 ($P < 0.0001$) and GSE78220 ($P = 0.0031$) than that of low risk patients with good prediction performance (Figs. 7A–7C). The proportion of PD or PD/SD was significantly increased in high risk group. Besides, patients with PD or PD/SD also exhibited higher risk score (Figs. 7D–7E).

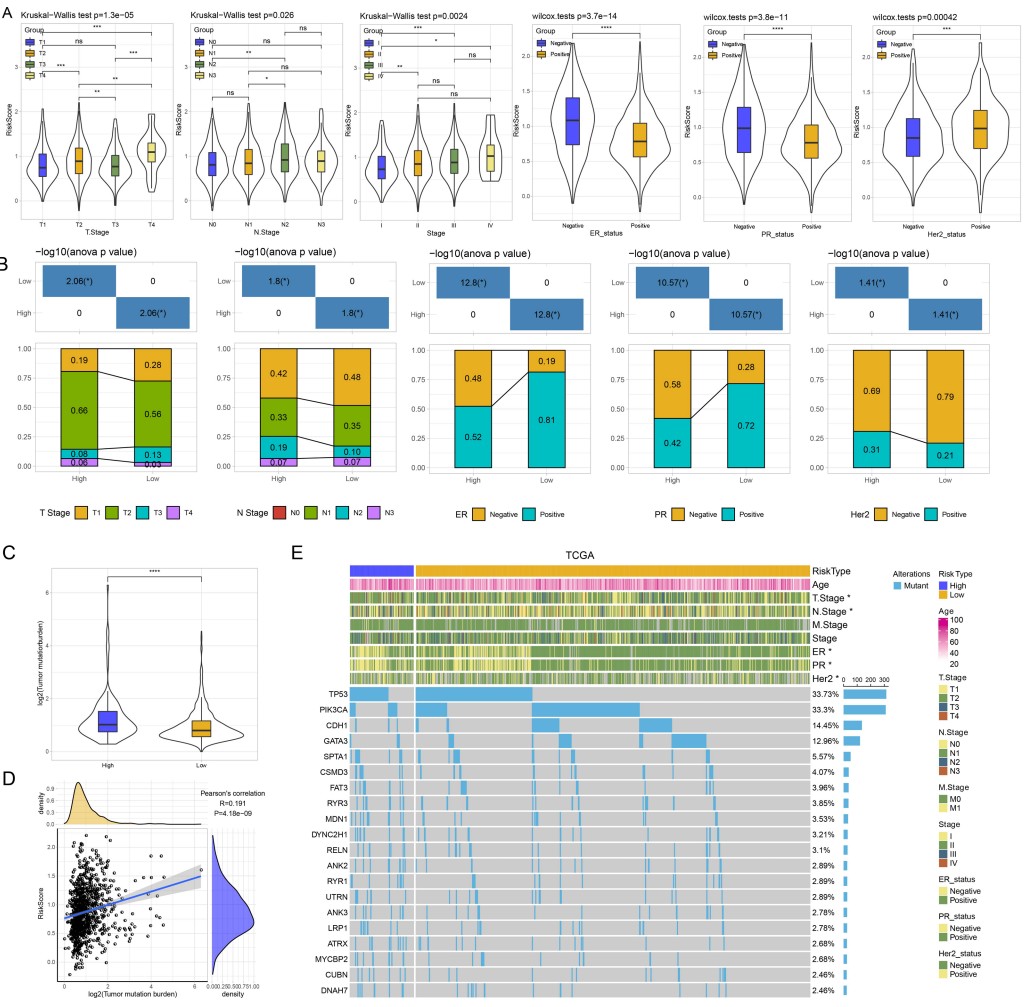

**Figure 4** (A) GSEA analysis between risk groups in TCGA and GSE datasets. (B–C) Representative results of GSEA analysis in GSE datasets and TCGA dataset.

## Construction of nomogram

Based on the decision tree, we determined six risk subtypes (Fig. 8A). Figure 8B displayed the different survival probability among six risk subtypes, where patients with subtype C3 were high risk patients while patients with subtype C1 and C2 were low risk patients (Figs. 8C–8D). Figures 8E–8F revealed that risk type was the most important factor related to prognosis. Furthermore, we constructed a nomogram and found that risk score had the greatest impact on survival (Fig. 8G). To evaluate the accuracy of this model, calibration curves for 1, 3, and 5-years were generated (Fig. 8H). The observed curve was highly consistent with predicted curve. Figure 8I displayed nomogram and risk score had the most powerful survival prediction ability.

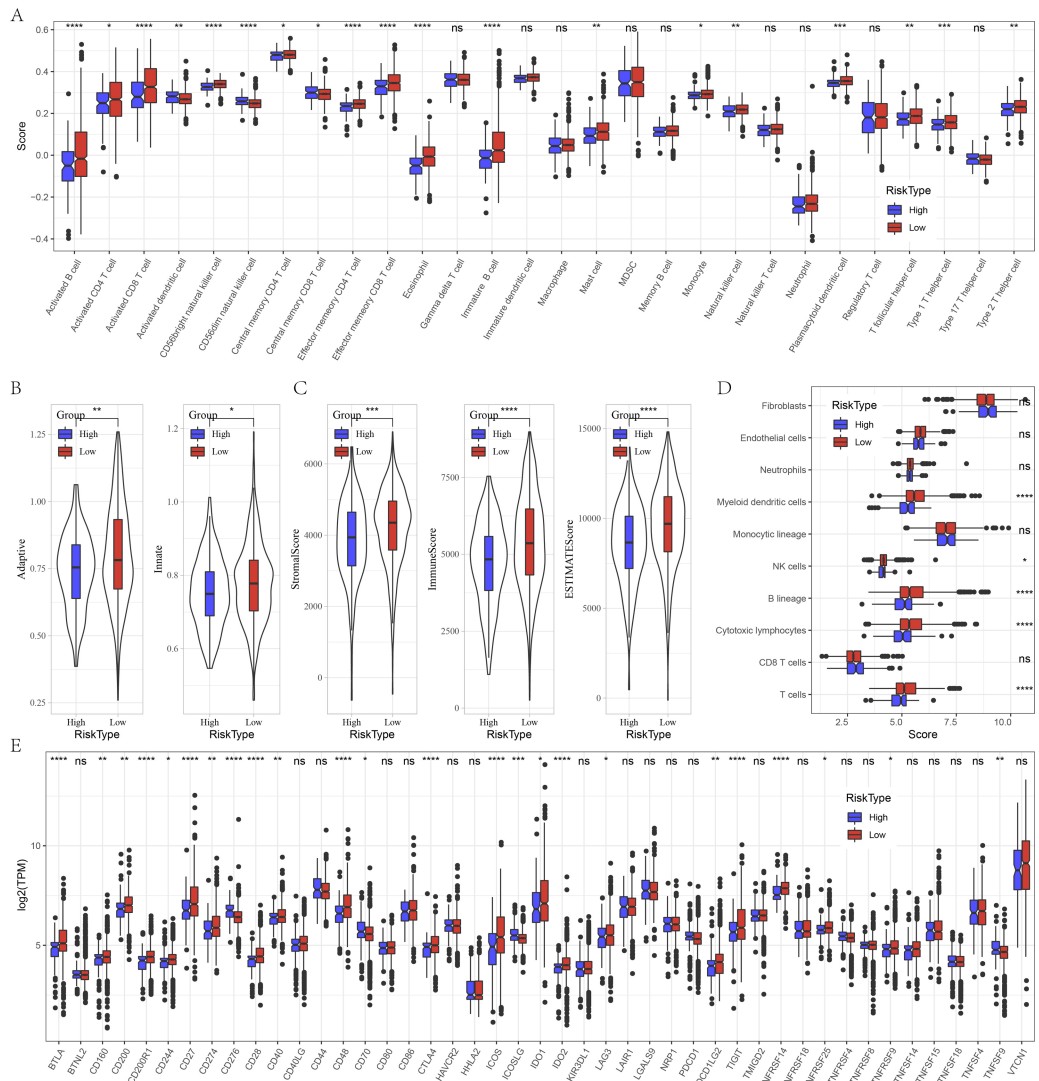

**Figure 5   Immune characteristics analysis between risk groups in GSE datasets.** (A) Box plots of 28 immune cells in high and low risk groups. (B) Differences of adaptive score and innate score between risk groups. (C) Comparison of immune scores predicted by ESTIMATE between risk groups. (D) MCP-counter score in high and low risk groups. (E) Box plots of immune checkpoints in high and low risk groups. ns represents $P > 0.05$; $*P < 0.05$, $**P < 0.01$, $***P < 0.001$, and $****P < 0.0001$.

## The cell viability, migration and invasion of breast cancer cells were promoted by circHIPK3

We detected the expression of circHIPK3 in MCF-7 cells by qRT-PCR, and we found that the mRNA expression of circHIPK3 was increased in MCF-7 cells in comparison to normal MCF10A cells (Fig. 9A). We subsequently examined the cell viability of MCF-7 and MDA-MB-231 after circHIPK3 inhibition, and the results showed that after circHIPK3 inhibition, the cell viability of these two kinds of breast cancer cell lines viability was significantly suppressed after circHIPK3 inhibition (Figs. 9B, 9C). We then test the

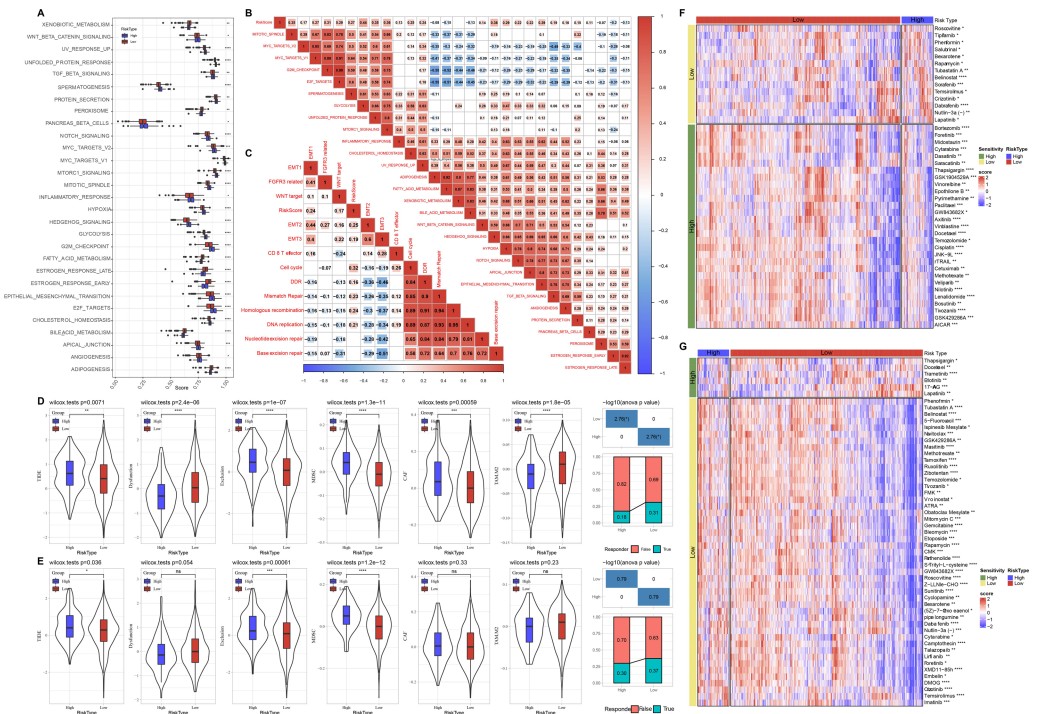

**Figure 6** **Pathway characteristics analysis and prediction of response to immunotherapy/chemotherapy of risk groups.** (A) Differences of pathway scores between risk groups in TCGA dataset. (B) Correlation analysis of risk score and pathway score in TCGA dataset. (C) Correlation analysis of risk score and human gene signatures in TCGA dataset. (D) Distribution of TIDE score between risk groups in TCGA dataset. (E) Distribution of TIDE score between risk groups in GSE datasets. (F) Visualization of distribution of 43 anti-tumor drugs between risk groups in TCGA dataset. (G) Visualization of distribution of 53 anti-tumor drugs between risk groups in GSE dataset. ns represents $P > 0.05$; *$P < 0.05$, **$P < 0.01$, ***$P < 0.001$, and ****$P < 0.0001$.

invasion and migration of MCF-7 (Figs. 9D–9F) and MDA-MB-23 (Figs. 9G–9I) cells after circHIPK3 inhibition, and we observed that the invasion and migration ability of the two cell lines was significantly diminished after circHIPK3 inhibition.

## DISCUSSION

It has been shown that the ceRNAs have regulatory roles in post-transcription of cancer cells. Recently, a circRNA-based ceRNA regulatory network has been established to classify breast cancer patients and facilitate the diagnosis, therapy and prognosis (*Zhong et al., 2022*). In breast cancer, miRNA signature has been developed to predict neoadjuvant chemotherapy responsiveness (*Xing et al., 2021*). In this study, we constructed a circHIPK3-based circRNA-miRNA-mRNA ceRNA network, and then established an 11 mRNA-based signature, which could predict the prognosis and anti-tumor response in BRCA carriers.

Investigation of the crucial roles of novel mRNAs in circRNA-miRNA-mRNA regulatory network may decipher the mechanism of mRNAs in breast cancer and has the potential to predict response to anti-tumor therapies. Small extracellular vesicles that contains ANGPTL2 derived from vascular endothelial cells has been reported to promote the
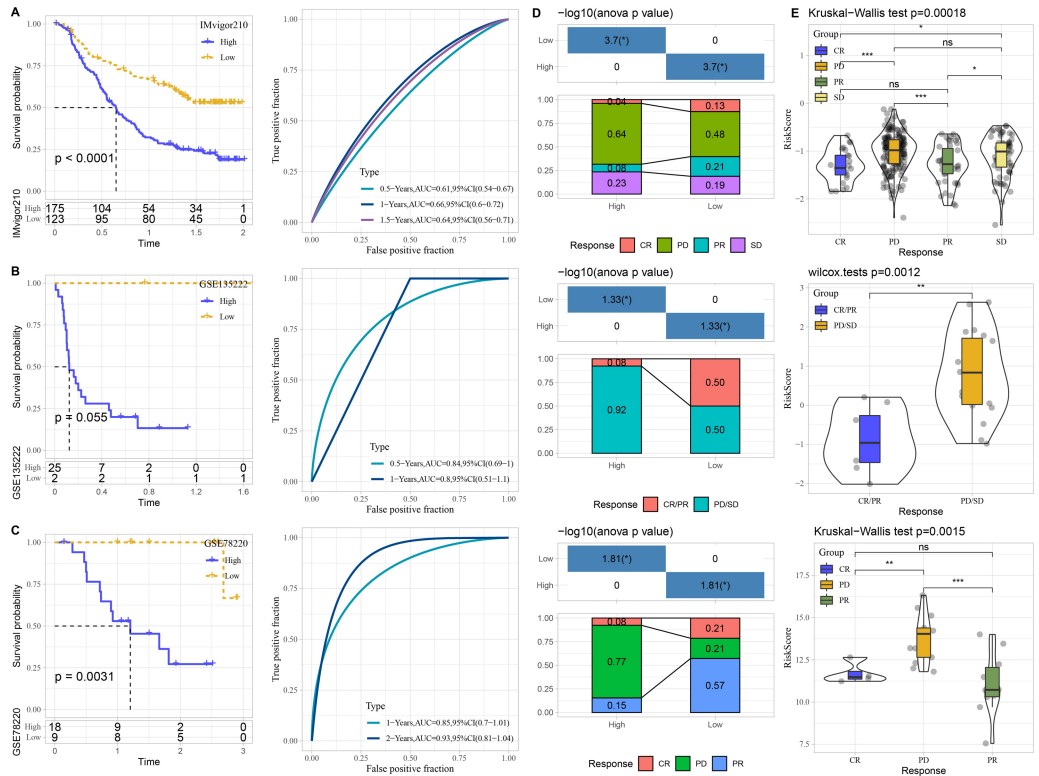

**Figure 7  Performance of predictive model in immunotherapy datasets.** Kaplan–Meier curves with ROC curves of risk score in IMvigor210. (A) dataset, GSE135222 (B) dataset, and GSE78220 (C) dataset. (D) Distribution of clinical response in high and low risk groups. (E) Comparison of risk score in patients with different clinical response. ns represents $P > 0.05$; $*P < 0.05$, $**P < 0.01$, $***P < 0.001$, and $****P < 0.0001$.

progression of leukemia (*Huang et al., 2021a*). A recent study has revealed that miR-378a-3p enhances osteolysis and then promotes release of ANGPTL2, which leads to tumor progression (*Wang et al., 2022*). CD1E is one type of CD1 molecule that is expressed in dendritic cells, and it has been identified to be associated with diffuse large B-cell lymphoma patients' long-term survival (*Gao et al., 2020*). LncRNA ENST00000582120 involves in epithelial-mesenchymal transition (EMT) through the regulation of COLEC12 (*Zhang et al., 2017*). COLEC12 acts as an innate immune-related risk mRNA that is related to immunosuppression in hepatocellular carcinoma (HCC) (*Huang et al., 2021b*). FAM117B is highly associated with bone formation and contribute to the metastasis of breast cancer to bone (*Sui et al., 2022*). MiRNA-876-3p exerts inhibitory effects on pancreatic adenocarcinoma cell growth and metastasis through regulating JAG2 (*Yang et al., 2018*). L1CAM is a promotor for tumor invasion, whereas inhibition of L1CAM can suppress breast cancer aggressiveness and attenuate cisplatin resistance *via* modulating AKT signaling (*Li et al., 2022*; *Zhang, Shen & Guo, 2022b*). Additionally, miRNA-296-3p could inhibit HCC cell proliferation and invasion targeting MSL2 and functions as a tumor suppressor (*Li, An & Gao, 2021*). PSME4 enhances aggressiveness of HCC *via* activating

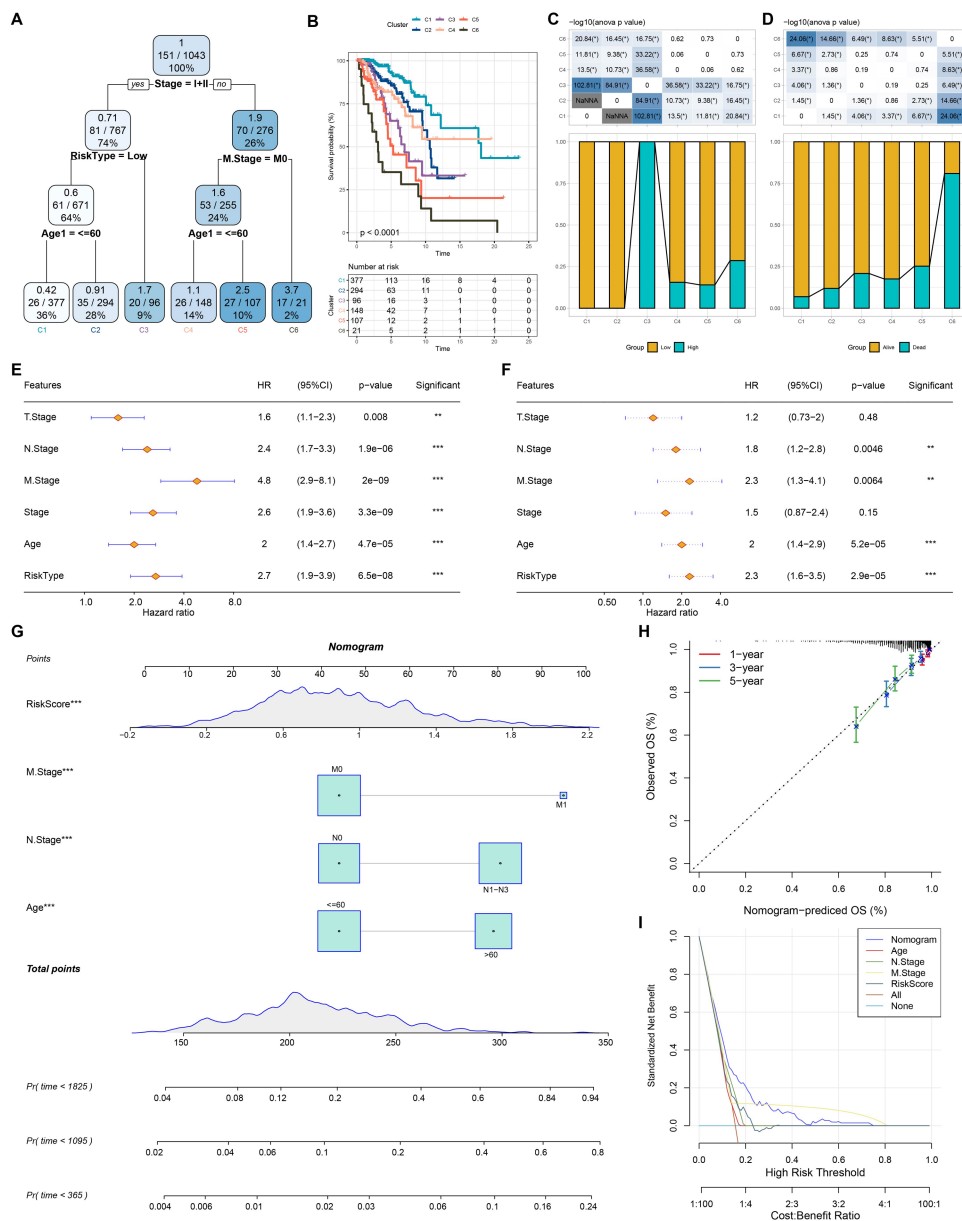

**Figure 8** **Construction of nomogram.** (A) For risk stratification optimization, patients with full-scale annotations including age, N stage, T stage, RiskType, Stage, and M stage were applied for developing a survival decision tree. (B) Significant differences of BRCA mutation patients' prognosis among the six risk subgroups. (C–D) Distribution of patients with risk and survival status among the six risk subgroups. (E–F) Univariate and multivariate Cox regression analysis were further used to select predictor of prognosis. (G) Construction of nomogram using risk type and clinicopathological features (T stage, N stage, M stage, Stage and age). (H–I) Calibration curves and DCA were further utilized to assess the accuracy and reliability of predictive risk model. $**p < 0.01$, $***p < 0.001$.

mTOR signaling (*Ge et al., 2022*). It has been documented that SH2B3 contributes to glioblastoma progression by transducing IL-6/gp130/STAT3 signaling (*Cai et al., 2021*). TTC39C has been screened as an alternative splicing-related mRNA that is implicated in

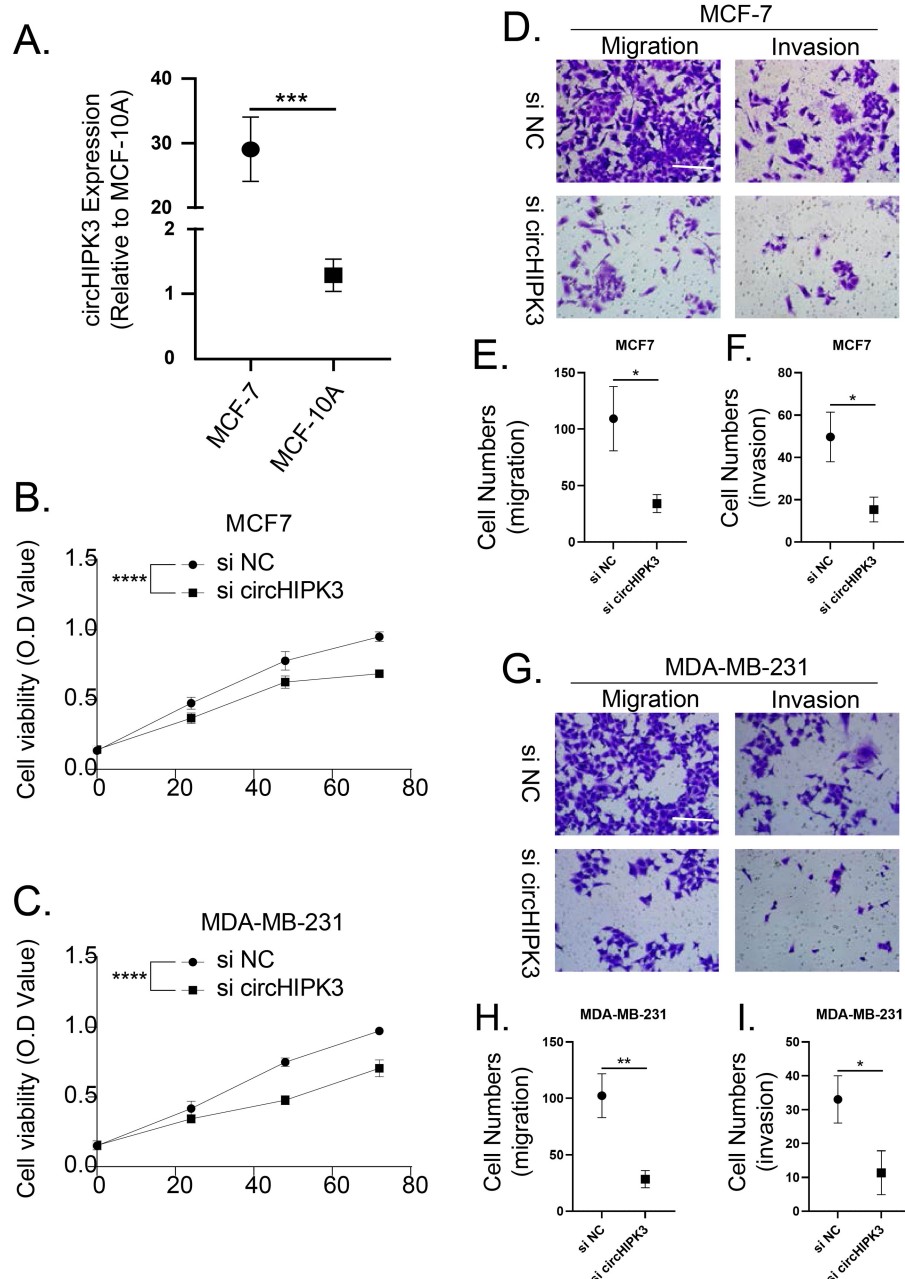

**Figure 9** **CircHIPK3 promoted breast cancer cells viability, invasion and migration.** (A) circHIPK3 was higher expression in breast cancer cells MCF-7. (B–C) si-circHIPK3 suppressed the cell viability of MCF-7 and MDA-MB-231 cells. (D–F) si-circHIPK3 suppressed the cell invasion and migration of MCF-7 cells. (G–I) si-circHIPK3 suppressed the cell invasion and migration of MDA-MB-231 cells. $^*p < 0.05$, $^{**}p < 0.01$, $^{***}p < 0.001$, $^{****}p < 0.0001$.

tumor microenvironment and immune infiltration of BRCA mutation patients (*Zhang et al., 2022a*). LncRNA SNHG8 promotes breast cancer progression and sponged miR-634, leading to the upregulation of ZBTB20 (*Xu et al., 2021*). The 11 mRNA-based signature

regulated by circHIPK3 might be closely related to breast cancer and might be promising therapeutic targets for BRCA carriers.

TP53, PIK3CA, and CDH1 were the most common genomic alterations for patients with breast cancer. CDH1 and PIK3CA mutation relate to metastasis in breast cancer patients (*Aftimos et al., 2021*). Additionally, the transfer of luminal A/B to Her2-enriched has a strong association with TP53 and/or PIK3CA mutation (*Aftimos et al., 2021*). TP53 mutation is significantly observed in Her2-, Her2+, and triple-negative breast cancer patients, and it is related to the dismal prognosis (*Meric-Bernstam et al., 2018*). The co-mutation of CDH1 with PIK3CA and ERBB2 contributes to endocrine resistance in breast cancer patients with invasive lobular carcinoma (*Davis et al., 2022*). Hence, evaluation of genomic alterations may reveal the pathogenesis, and help histologic classification and personalized treatment for patients with breast cancer. In the present study, we found high risk patients exhibited higher proportion of T2 stage, N2 stage, positive ER, negative PR, and positive Her2 and possessed highly mutated TP53, PIK3CA, and CDH1. The established risk score may provide a supplement for classification of breast patients and guide precision treatment.

As a crucial microenvironmental factor, hypoxia facilitates tumor cells survival and promotes tumor growth and metastasis. Tumor cells prefer glycolysis as the source of ATP under hypoxia (*Paredes, Williams & Martin, 2021*). Glycolysis products metabolic intermediates that is essential for tumor cell growth and adhesion. Intriguingly, compelling evidence has confirmed that hypoxia-driven activation of HIF-1α enhances EMT, which involves in tumor metastasis, immune escape and drug resistance (*Saxena, Jolly & Balamurugan, 2020*). It has been found that HYPOXIA, GLYCOLYSIS, and EPITHELIAL_MESENCHYMAL_TRANSITION were significantly enriched in the high risk group, which might explain the poor prognosis of high risk patients. Moreover, we also found less immune cell infiltration in tumors of high patients whereas more infiltration of NK cells, cytotoxic lymphocytes and T cells in low risk patients. This is an important reason for unfavorable prognosis of high risk patients.

Although we have constructed a circHIPK3-based ceRNA network, the regulatory mechanism of the important circRNA, miRNAs and mRNAs within the ceRNA network need further exploit. The predictive value of this mRNA signature has been validated in TCGA dataset, while larger samples should be enrolled in further prospective studies to verify our retrospective data.

## CONCLUSION

In conclusion, we constructed a circHIPK3-based ceRNA regulatory network and identified a novel 11-mRNA-based signature in BRCA carriers. The current study might help understanding the underlying mechanism of mRNA signature and offer potential targets for prognosis and personalized treatment in breast cancer patients with BRCA mutation.

### Funding

The authors received no funding for this work.

### Competing Interests

The authors declare there are no competing interests.

### Author Contributions

- Qi-xin Lian conceived and designed the experiments, authored or reviewed drafts of the article, and approved the final draft.
- Yang Song conceived and designed the experiments, authored or reviewed drafts of the article, and approved the final draft.
- Lili Han performed the experiments, prepared figures and/or tables, and approved the final draft.
- Zunxian Wang performed the experiments, authored or reviewed drafts of the article, and approved the final draft.
- Yinhui Song analyzed the data, authored or reviewed drafts of the article, and approved the final draft.

### Data Availability

The data is available at OSF: Song, Yinhui. 2023. ''Data Source.'' OSF. May 22. doi: 10.17605/OSF.IO/DS6TA.

### Supplemental Information

Supplemental information for this article can be found online at http://dx.doi.org/10.7717/peerj.15572#supplemental-information.

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
