# Peer review of "Development of a circHIPK3-based ceRNA network and identification of mRNA signature in breast cancer patients harboring BRCA mutation"

_PeerJ, doi:10.7717/peerj.15572_

## Round 0.1 · original submission · Minor Revisions

Please revise your manuscript following the suggestions of two Reviewers.

Reviewer 1 ·

Basic reporting

no comment

Experimental design

no comment

Validity of the findings

no comment

Additional comments

CircHIPK3 could promote tumor growth, such as glioma, prostate cancer, epithelial ovarian cancer, BCRA and colorectal cancer. In this work, circHIPK3 has been focus on the breast cancer patient that carrying BRCA. Based on ceRNA (circRNA-miRNA-mRNA), a group of mRNA were detected that have strong relation to circHIPK3, what's more, 11 mRNAs were defined as the prognostic mRNAs and the risk model can divide patient into high and low group, ant it's robust in independent datasets. Two risk group have different characteristic, for example, high risk group exhibited higher proportion of T2 stage, N2 stage, positive ER, negative PR, and positive Her2.

1. What does it mean of " breast cancer patients with BRCA" at line 51, with BRCA dysregulation or mutation? 2. The detail function should be listed at line 40, and lowercase the first letter of "Transwell".
2. The cutoff should set at right form at line 126, P <0.05?.
3. The stepAIC at line 129 seems not correspond to the LASSO in the abstract part.
4. The cutoff for dividing patients into high or low risk group is not clear.
5. Whether mutation analysis in this work have the crosstalk to ceRNA network? The result should have further crosstalk analysis rather than simple exhibition of top mutate genes.
6. Keep the same bold font of figure at the last paragraph.
7. Too small font size in figure 6.
8. The GSE dataset was divided randomly into two Train and Test datasets with a 1:1 ratio, what datasets in Fig 2c-d? Test dataset and GSE datasets at line 245 seems the same dataset.

·

Basic reporting

This study identified circHIPK3 related mRNA through circHIPK3 related ceRNA network, and constructed the breast cancer prognosis risk score. This score can not only predict the prognosis of breast cancer patients, but also evaluate the immune response and drug sensitivity of breast cancer patients. This study is innovative, reasonable, and logical, but some results still need to be further optimized.
1. In the Method section of Abstract, the author should introduce relevant analysis methods, such as the analysis of differences between risk scores and clinical indicators, correlation analysis methods, survival analysis methods, etc.
2. In the introduction part, the author did not introduce the difficulties faced by the current breast cancer treatment, such as the current markers can not accurately define the personalized treatment of breast cancer, and the drug resistance in the process of drug treatment of breast cancer.
3. In Method, there is a lack of introduction to statistical methods and the use of related software, such as the R language version.

Experimental design

no comment

Validity of the findings

This validation was conducted using independent datasets and further validated through in vitro experiments. I have no further comments

Additional comments

1. In the "Construction of a circHIPK3-based ceRNA network" section, it is not enough to simply describe the analysis of the data. The findings can be inferred, such as circHIPK3 competing with related mRNA for miRNAs, which may play a regulatory role in the occurrence of cancer.
2. For the result "GSEA analysis between risk groups", it should be adjusted before "Analysis of clinicopathological features and SNV between risk groups". After completing the risk scoring and grouping, it is more appropriate to conduct a functional enrichment analysis.
3. The results of "Distribution of risk score in clinicopathological features" and "Analysis of clinicopathological features and SNV between risk groups" are somewhat similar, and the author should merge and adjust them appropriately.
4. Figure 2A should indicate the meaning represented by each column of values, such as HR, pvalue, etc. The colors of the high and low risk groups in the 2B-E diagram should be interchanged, with red indicating that the high risk group is more suitable.
5. The insignificant clinical features in Figure 8 can be chosen not to be displayed, such as age and M staging.

---

## Round 0.2 · accepted · Accept

I am satisfied with the revisions made by the authors.